**Data Availability Statement:** Data are available as Supporting information except for the data

# The impact of the Covid-19 pandemic on the incidence of diseases and the provision of primary care: A registry-based study

Steve Van den Bulck[1][*]*, Jonas Crèvecoeur[2,3][*]*, Bert Aertgeerts[1], Nicolas Delvaux[1], Thomas Neyens[2,3], Gijs Van Pottelbergh[1], Patrick Coursier[1], Bert Vaes[1]

**1** Department of Public Health and Primary Care, Academic Center for General Practice, KU Leuven, Leuven, Belgium, **2** I-BioStat, Data Science Institute, Hasselt University, Hasselt, Belgium, **3** Department of Public Health and Primary Care, I-BioStat, Faculty of Medicine, KU Leuven, Leuven, Belgium

☯ These authors contributed equally to this work.
* Steve.vandenbulck@kuleuven.be (SVdB); jonas.crevecoeur@kuleuven.be (JC)

## Abstract

### Introduction

The Covid-19 pandemic had a tremendous impact on healthcare but uncertainty remains about the extent to which primary care provision was affected. Therefore, this paper aims to assess the impact on primary care provision and the evolution of the incidence of disease during the first year of the Covid-19 pandemic in Flanders (Belgium).

### Methods

Care provision was defined as the number of new entries added to a patient's medical history. Pre-pandemic care provision (February 1, 2018–January 31, 2020) was compared with care provision during the pandemic (February 1, 2020-January 31, 2021). A large morbidity registry (Intego) was used. Regression models compared the effect of demographic characteristics on care provision and on acute and chronic diagnoses incidence both prior and during the pandemic.

### Results

During the first year of the Covid-19 pandemic, overall care provision increased with 9.1% (95%CI 8.5%;9.6%). There was an increase in acute diagnoses of 5.1% (95%CI 4.2%;6.0%) and a decrease in the selected chronic diagnoses of 12.8% (95% CI 7.0%;18.4%). Obesity was an exception with an overall incidence increase. The pandemic led to strong fluctuations in care provision that were not the same for all types of care and all demographic groups in Flanders. Relative to other groups in the population, the pandemic caused a reduction in care provision for children aged 0–17 year and patients from a lower socio-economic situation.

underlying Fig 2. The minimal data set necessary to validate this figure (analysis) contains identifying patient-level data which cannot be suitably de-identified or aggregated and can therefore only be accessed inside a monitored analysis environment. These restrictions were imposed by the Belgian National Information Security Committee: section Social Security and Health. Data requests may be sent to Mr. Roel Heijlen, Data Protection Officer healthdata.be (Sciensano), Roel. Heijlen@sciensano.be, Rue Juliette Wytsmanstraat 14, 1050 Brussels. Interested researchers will need to provide their name, first name, professional organization name, email address, mobile number and the database of interest to request access.

**Funding:** TN gratefully acknowledges funding by the Internal Funds KU Leuven (project number 3M190682).

**Competing interests:** The authors have declared that no competing interests exist.

## Conclusion

This paper strengthened the claim that Covid-19 should be considered as a syndemic instead of a pandemic. During the first Covid-19 year, overall care provision and the incidence of acute diagnoses increased, whereas chronic diseases' incidence decreased, except for obesity diagnoses which increased. More granular, care provision and chronic diseases' incidence decreased during the lockdowns, especially for people with a lower socio-economic status. After the lockdowns they both returned to baseline.

## Introduction

During the last year, the Covid-19 pandemic had a tremendous impact on many aspects of society and especially on healthcare [1]. At the start of the pandemic and subsequent lockdowns, primary care and hospitals were flooded with Covid-19 patients and the provision of regular care was forcibly reduced to a strict minimum [2, 3]. This was certainly the case for many countries in Europe, including Belgium [4].

The impact on healthcare in general and on the care for patients with chronic diseases in particular, still remains largely unknown, but the collateral damage is suspected to be high [5, 6]. Regular care for chronic diseases such as hypertension, diabetes and chronic obstructive pulmonary disease (COPD) was affected the most in various nations worldwide [7]. In addition, there are indications that patients were postponing many different types of care, such as prescription refills and imaging investigations [8, 9]. Especially people with a lower socio-economic status could be prone to this because the Covid-19 pandemic and the subsequent preventive measures can exacerbate already existing vulnerabilities and increase inequality [10–12]. In this context, the Covid-19 pandemic is also called a syndemic by other authors [13]. The concept syndemic is used to label the synergistic interaction of two or more coexistent conditions, biological, economical and/or environmental factors and the resulting excess burden of disease [14, 15].

It remains unclear to what extent primary care provision was influenced and which patients were most affected, but the scarce available evidence stresses the need to monitor diseases and social determinants for operational and population management [12, 16–18]. To evaluate the extent of delaying care, the registration of disease incidences in the electronic health record (EHR) of the general physician (GP) can provide valuable insights. In addition, the EHR of the GP also offers a wealth of data on healthcare utilization such as imaging, medication and physiotherapy orders that can be used to investigate the impact of the different lockdowns on the primary care provision.

Large primary care data repositories are available that routinely collect healthcare data that can be used for the aforementioned monitoring purposes [19–24]. However, there is a lack of research concerning the use of these registries to evaluate the impact of the Covid-19 pandemic on diagnosing diseases and on evaluating the provision of primary care during the pandemic, especially for patients with a lower socio-economic status. In addition, qualitative research indicated that there was a disturbance in the delivery of chronic care in Belgium during the pandemic, although there is little quantitative research to support these findings [6]. In order to substantiate the above with more generalizable data, it is crucial to examine the evolution of (chronic) diseases (more specifically their incidences during the Covid-19 pandemic), study

primary care provision during the Covid-19 pandemic and assess whether social determinants influenced these measures.

Therefore, we aim to assess the impact of Covid-19 measures on primary care provision using a large morbidity registry, and more particularly how often and by whom primary care was consulted during the first year of the Covid-19 pandemic. In addition, we aim to evaluate the evolution of disease incidences in Flanders (Belgium) during the first year of the Covid-19 pandemic.

## Material and methods

### Data

Data were used from Intego, a registry network collecting detailed medical information on patients from general practices in Flanders, i.e. the Dutch speaking region of Belgium [25]. In order to use Intego data for incidence (and prevalence) analyses, an algorithm developed by the Dutch NIVEL institute was used [26]. The study population consisted of 397,489 distinct patients who visited one of the 105 Intego primary care practices between February 1, 2018 and January 31, 2021. Data were automatically collected via the EHR of the participating practices and included patient characteristics, medical history and socio-economic status of a patient. Available demographic characteristics included year of birth, sex, nationality and the place of residence of the patient (postal code). The medical history consisted of diagnoses coded using the International Classification for Primary Care (ICPC-2), laboratory tests, drug prescriptions, physical therapy referrals or radiology orders. Patients with a low socio-economic status were identified based on whether they are eligible for increased reimbursements under the national healthcare insurance policy.

**Care provision.** We define care provision as the number of new entries added to a patient's medical history over a predefined time window. These entries included medication prescriptions, lab tests, measurements of medical parameters, diagnoses, physical therapy referrals and radiology orders. In the computation of care provision, multiple lab tests or medication prescriptions registered on the same date were counted as one. Care provision includes any action by the GP resulting in a change in the patient's medical history. This includes face-to-face contact, telemedicine and prescription refills. For a more granular view, we split care provision into separate metrics per record type (diagnoses, lab test, prescription, ...).

**Intervention period.** The impact of the Covid-19 pandemic was evaluated by comparing care metrics prior to the pandemic (February 1, 2018 –January 31, 2020) and after the start of the pandemic (February 1, 2020, January 31, 2021). In this paper, we refer to the second period as the first year of the Covid-19 pandemic. Within the first year of the Covid-19 pandemic we labeled the periods March 1, 2020 –May 30, 2020 and October 1, 2020 –November 30, 2020 as the first and second wave of the pandemic, respectively.

**Diagnoses and medication.** For this study, we distinguished between chronic and acute diagnoses based on the registered ICPC-2 code, see S1 Appendix. The attention values registered for each diagnosis in the EHR (active, passive relevant, passive non-relevant and not present) were used to omit diagnoses with code 'not present'. The aforementioned algorithm, designed by the Dutch NIVEL, enabled a differentiation between new cases and chronic cases from follow-up treatment for the same disease in the same patient [26]. We further split the chronic diagnoses in small subgroups corresponding to diabetes, chronic renal problems, chronic lung problems, chronic liver problems, chronic cardiovascular problems, obesity, chronic neurological problems, and cancer. When prescribing medication, GPs indicated whether the treatment was chronic. This criterion was used to distinguish between chronic and acute medication. We identified Covid-19 related lab tests and diagnoses based on a free

text field description available in the data. Descriptions containing one of the words 'covid', 'corona', or 'sars' were interpreted as Covid-19 related.

## Statistical analysis

Relative care provision was computed as the ratio between the care provision in the first year of the Covid-19 pandemic and the provision during the previous two years. More granular, bi-weekly care data were obtained by comparing the same calendar weeks within both periods.

Generalized linear models analyzed care provision per patient per year as a function of the covariates age, sex, nationality, living in a center city and socio-economic status with a reference category (aged 18–35, female, Belgian, living outside center cities, high socio-economic status). Besides these demographic covariates, we include the practice visited in the model to correct for registration differences across practices. Fitted practice-specific effects are included in S2 Appendix, but are not further interpreted. We chose a fixed practice effect over a random effect as the distribution of these effects is strongly non-normal and the high number of patients per practice allows for an accurate estimation. We modeled care provision using a Poisson distribution with log link. The model estimated the change in care provision for the reference category in the first year of the Covid-19 pandemic and the multiplicative effect of demographic covariates on care provision both in the period before the pandemic as during the first year of the pandemic. This approach allowed us to assess the overall impact of the pandemic as well as how this impact differs for various demographic groups in the population.

The analysis was repeated for the number of acute diagnoses per patient and for the number of chronic diagnoses from the subgroups defined in S2 Appendix. Statistical analysis was conducted in R version 4.0.3.

## Ethical approval

The Intego procedures were approved by the KU Leuven Ethics Committee (nr. ML1723) and by the National Privacy Commission's Sectoral Committee (decision nr. 13.026 of March 19, 2013, last modified on April 17, 2018). Written informed consent was obtained from every of the 105 Intego primary care practices.

## Results

### Evolution of care provision

During the first year of the Covid-19 pandemic, the reference category (aged 18–35, female, Belgian, living outside center cities, high socio-economic status) registered an increase in care provision of 9.1% (95% CI 8.5%-9.6%), an increase in acute diagnoses of 5.1% (95% CI 4.2%-6.0%) and a decrease in the selected chronic diagnoses of 12.8% (95% CI 7.0%-18.4%). However, these overall changes in care during the first year of the Covid-19 pandemic varied largely across specific periods.

During the first wave of the pandemic care provision was strongly reduced with observed peak reductions in radiology orders (-74%), lab tests (-50%), chronic diagnoses (-30%) and acute diagnoses (-28%) (Fig 1). The decrease in acute diagnoses was associated with a similar decrease in acute medication prescriptions, whereas chronic medication prescriptions remained the same in spite of the decrease in new chronic diagnoses. The number of physical therapy referrals also remained the same.

In between the first and second Covid-19 wave, acute diagnoses (excluding Covid-19 related diagnoses) returned to pre-pandemic levels. Around 25% of the acute diagnoses in this period were Covid-19 related. The number of chronic diagnoses and radiology orders slightly

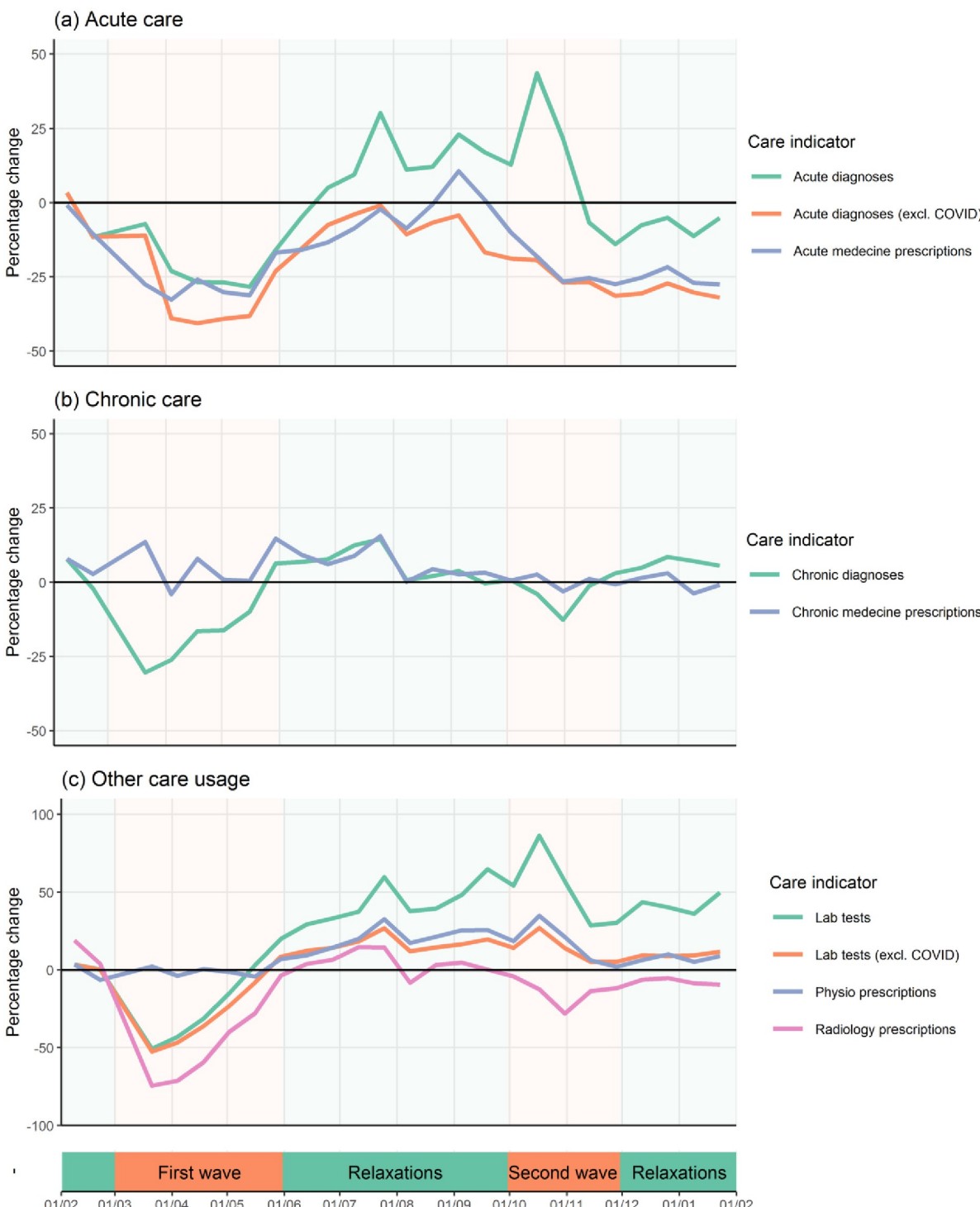

**Fig 1. Evolution of care provision for various care indicators in the first year of the Covid-19 pandemic (February 1, 2020–January 31, 2021) relative to the two previous years.**

**Table 1. Average number of diagnoses per 1,000 patient years during the first year of the Corona pandemic and percentage change in diagnose incidence relative to the previous two years along with 95% confidence intervals.** Results are stratified by calendar period and diagnoses group.

| Chronic condition | March-May | June-September | October-November | December-January | February-January |
|---|---|---|---|---|---|
| | First wave | Relaxations | Second wave | Relaxations | Full year |
| diabetes | 4.32 (-25% CI -34,-15) | 6.36 (+16% CI +5,+28) | 6.06 (+2% CI -10,+18) | 7.39 (+3% CI -8,+17) | 5.91 (+0% CI -5,+6) |
| renal | 2.75 (-9% CI -23,+6) | 2.90 (-6% CI -18,+7) | 2.82 (-17% CI -32,+0) | 3.30 (-10% CI -25,+7) | 2.92 (-8% CI -15,-1) |
| lung | 10.70 (-42% CI -47,-38) | 7.17 (-38% CI -43,-33) | 7.90 (-55% CI -60,-50) | 8.82 (-55% CI -60,-50) | 9.28 (-43% CI -45,-40) |
| liver | 2.56 (-22% CI -34,-8) | 3.52 (+7% CI -5,+22) | 3.49 (-10% CI -25,+6) | 3.87 (-3% CI -18,+14) | 3.33 (-5% CI -12,+2) |
| cardiovascular | 17.56 (-24% CI -29,-18) | 20.23 (+11% CI +5,+18) | 23.68 (+0% CI -7,+6) | 27.95 (+12% CI +5,+21) | 21.02 (-1% CI -4,+1) |
| obesity | 3.21 (-40% CI -48,-31) | 5.86 (+18% CI +7,+31) | 5.34 (+11% CI -4,+28) | 9.25 (+80% CI +59,+104) | 5.59 (+10 CI +3%,+16) |
| neurological | 3.26 (-29% CI -39,-19) | 4.53 (+2% CI -8,+15) | 4.57 (-15% CI -27,-1) | 4.51 (-3% CI -17,+13) | 4.16 (-10% CI -16,-4) |
| cancer | 5.01 (-23% CI -31,-13) | 7.47 (+18% CI +7,+29) | 7.53 (+1% CI -10,+14) | 7.14 (+2% CI -9,+16) | 6.70 (+0% CI -4,+6) |

increased. Lab tests strongly increased with around 50% more lab tests per patient compared to the previous two years. A large part of this increase, although not completely, was caused by Covid-19 tests.

The second wave was characterized by a strong increase in the number of Covid-19 diagnoses and lab tests ordered by GPs. After a short delay the increase in acute diagnoses was followed by a decrease in radiology orders and chronic diagnoses, but this decrease was smaller than during the first wave of the pandemic.

## Evolution of selected chronic diagnoses during the pandemic

Table 1 contains the incidence of selected chronic diagnoses per 1,000 patient years during the first year of the pandemic as well as the percentage change in diagnosis incidence relative to the previous two years. Overall, there was a significant drop in chronic diagnoses during the first wave of the pandemic. Only for renal problems the decrease in incidence was not significant, which could be the result of higher uncertainty on these estimates due to lower incidence rates. The decrease in incidence was the largest for chronic lung problems and obesity with 42% (95% CI 38%-47%) and 40% (95% CI 31%-48%), respectively.

During the summer months (June to September 2020) incidence rates for chronic diagnoses returned to pre-pandemic levels with a slight, but statistically significant increase in the number of diagnoses for diabetes, chronic cardiovascular problems, obesity and cancer. In contrast to other chronic conditions the number of chronic lung diagnoses remained low with 38% (95% CI 33%-43%) less diagnoses relative to previous years.

Most chronic diagnosis incidence rates decreased slightly during the second pandemic wave and returned to pre-pandemic levels during the months December-February 2021. Exceptions were chronic cardiovascular conditions and obesity for which we observed a rise of 12% and 80% in the months following the second wave of the pandemic and chronic lung conditions for which the incidence remained low throughout the whole year.

## Demographic effects on care provision and diagnoses

Fig 2 shows the multiplicative effect of the parameters of the generalized linear model for care provision along with 95% confidence intervals. Overall, care provision was higher for older patients, females and patients with a lower socio-economic status. For the reference category, care provision increased by 9.1% in the first year of the Covid-19 pandemic. The increase was more pronounced for patients living in center cities and less pronounced for children and patients with a low socio-economic status.

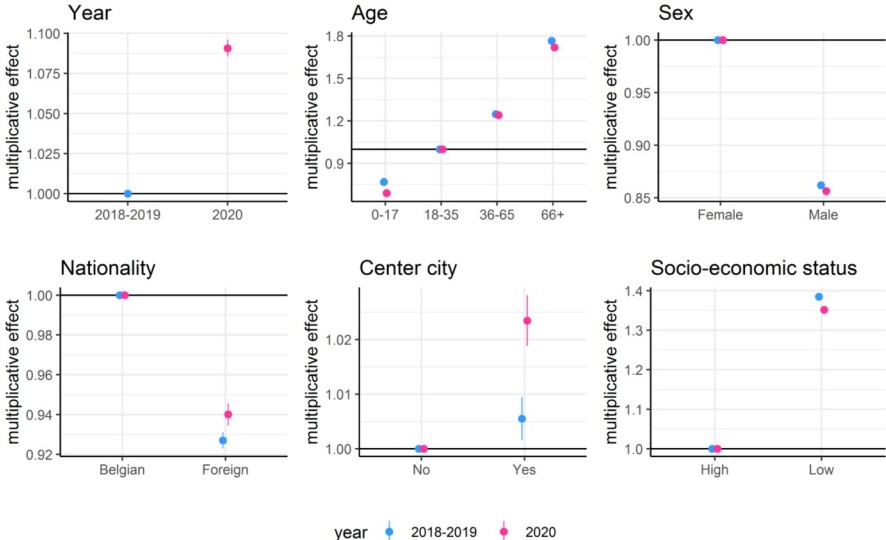

**Fig 2. Multiplicative effect and associated 95% confidence intervals of covariates on the expected care provision per patient per year.** Separate parameters were estimated for the period February 1, 2018 to January 31, 2020 and the period February 1, 2020 to January 31, 2021 which are labeled 2018–2019 and 2020, respectively.

In Fig 3 the multiplicative effects are shown for the number of chronic diagnoses. Notice that we only include the chronic diagnoses belonging to one of the subgroups defined in S2 Appendix. Chronic diagnoses were strongly age-related, with six times more recorded chronic diagnoses for patients aged 66 years and older as compared to those aged 18–35. Diagnoses were more common in males (+17% in 2018–2019) and patients with a lower socio-economic status (+44% in 2018–2019). In 2020, during the Covid-19 pandemic, the number of registered chronic diagnoses in the reference category decreased by 13% (95%CI 10%-16%), compared to 2018–2019. This decrease was slightly stronger for patients with a low socio-economic status.

Fig 4 shows the effect of demographic covariates on acute diagnoses. In regular times the number of acute diagnosis is higher for patients aged 18–35, females, patients with a Belgian nationality and patients having a lower socio-economic status. Overall, the number of acute diagnoses per patient year increased during the Corona pandemic, but this increase was not equally spread over all demographic groups. Relative to those aged 18–35, children received significantly less acute care during the Corona pandemic as compared to previous years. More acute care was given to patients living in center cities and to those having a foreign nationality.

## Discussion

### Main findings

This registry-based study evaluated the impact of the Covid-19 pandemic and subsequent lock-down(s) on primary care provision and the evolution of disease incidence in Flanders (Belgium) by comparing the first year of the Covid-19 pandemic with the two previous years. During the first year of the Covid-19 pandemic overall care provision and the incidence of acute diagnoses increased slightly due to Covid-19 related care, whereas the incidence of chronic diseases in general was considerably reduced in the reference group. Obesity was the only chronic disease of which the incidence increased during the first Covid-19 year. In contrast with the general evolution, children aged 0–17 had less acute diagnoses during the Covid-19 pandemic. This is possibly the result of school closures, resulting in less infections in

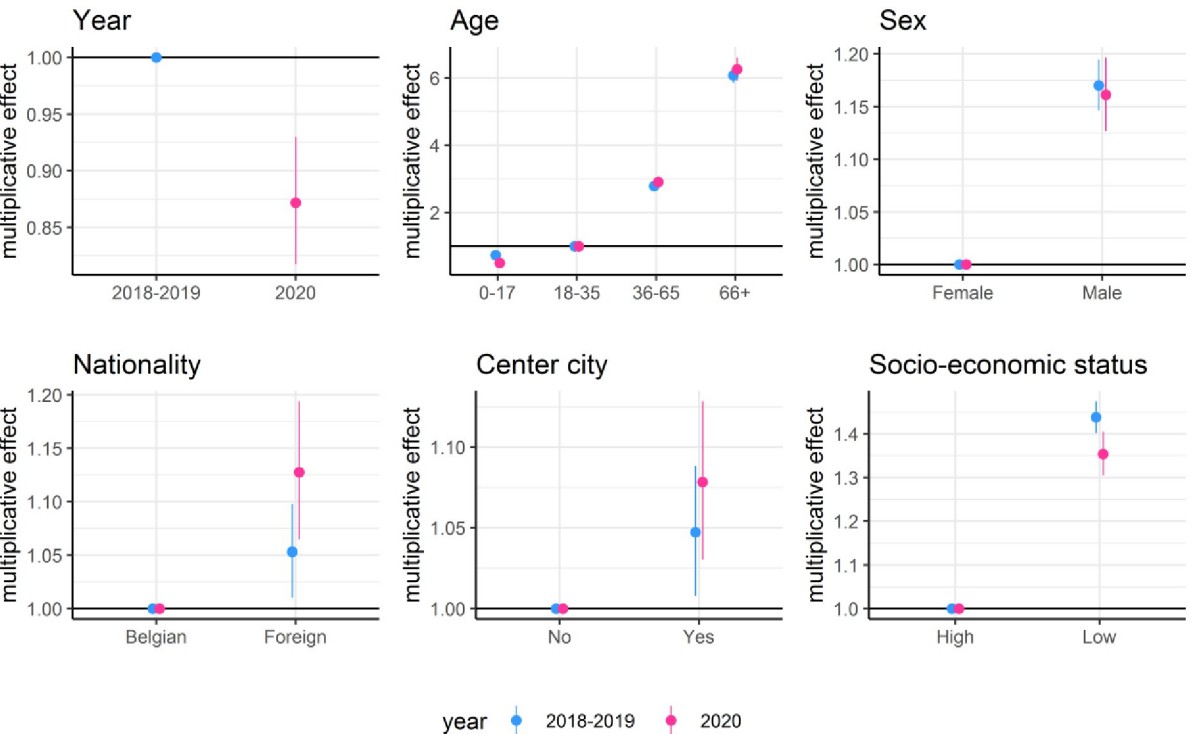

**Fig 3. Multiplicative effect and associated 95% confidence intervals of covariates on the expected number of chronic diagnoses per patient per year.** Separate parameters were estimated for the period February 1, 2018 to January 31, 2020 and the period February 1, 2020 to January 31, 2021 which are labeled 2018–2019 and 2020, respectively.

general, and of children receiving less Covid-19 related treatments and tests as they were less sick when they became infected [27]. On a more granular level, there was a drop in care provision as well as chronic diseases' incidences during the lockdowns for people with a lower socio-economic status relative to those with normal and high socio-economic status.

This latest finding reinforced the claim of other authors that the Covid-19 pandemic should rather be considered as a Covid-19 syndemic [13]. Describing Covid-19 as a syndemic corresponded with previous work that identified different socio-economic factors influencing the variation in mortality and morbidity of Covid-19 [28]. For example, ethnic minority groups with a low income in England and Wales, had a greater chance of being exposed to the virus, to die from it, and to transmit the disease to relatives living in the same house [29]. However, other authors have argued that Covid-19 should not be called a global syndemic, as the different factors that drive the disease are contextual [30]. For instance, the management of Covid-19 differed greatly around the world, and while in the USA the disease could be called syndemic this was not the case in New-Zealand [30]. Our results indicated that In Flanders (Belgium) Covid-19 could be consider syndemic and illustrated that in the management of both Covid-19 and chronic diseases, attention should be payed to social determinants of health, even in a Western country with a well-established social security system [12]. Moreover, these social determinants should also play an important role in the timely access to telemedicine since people with a lower socio-economic status could be hindered by insufficient digital literacy, especially during the Covid-19 syndemic [31, 32].

Care provision was assessed by investigating different care metrics for acute care, chronic care and other care. Despite the overall increase in care provision during the first Covid-19 year, our findings indicated on a more detailed level that all of these metrics were reduced

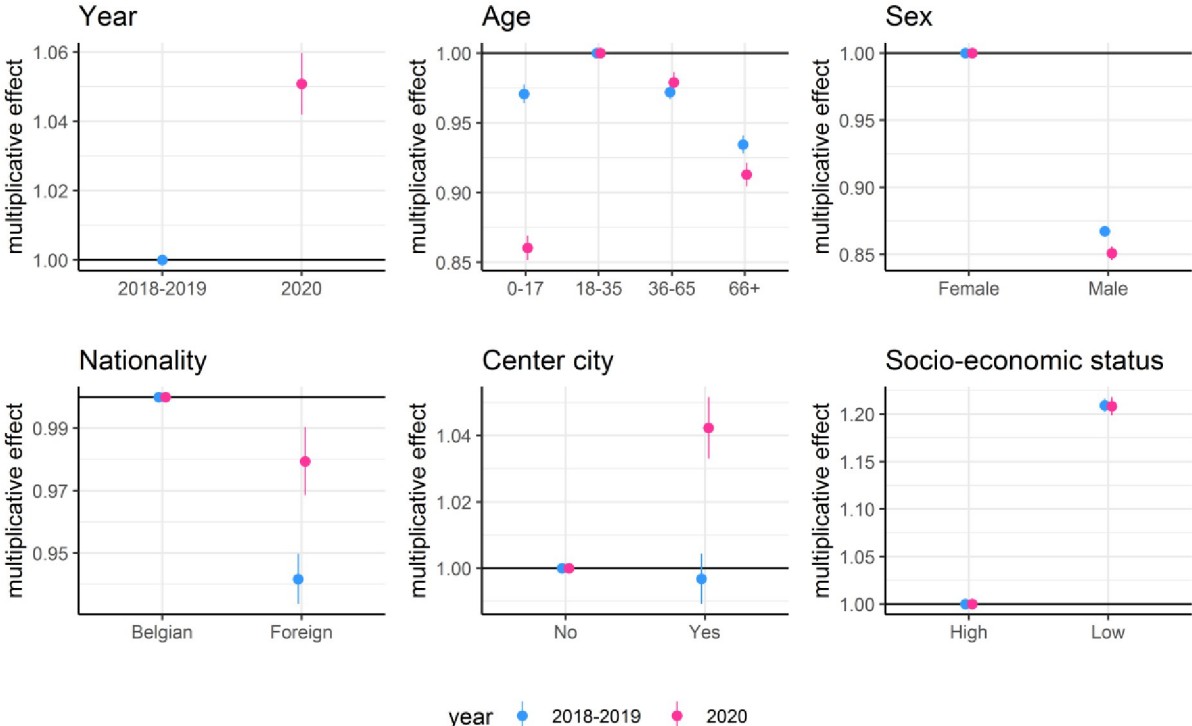

**Fig 4. Multiplicative effect and associated 95% confidence intervals of covariates on the expected number of acute diagnoses per patient per year.** Separate parameters were estimated for the period February 1, 2018 to January 31, 2020 and the period February 1, 2020 to January 31, 2021 which are labeled 2018–2019 and 2020, respectively.

during the first wave of Covid-19, except for chronic medication prescriptions and physical therapy referrals. These observations showed a decrease in the use of primary healthcare in Flanders (Belgium) during the lockdowns as was also the case in other countries [2, 7, 9, 33]. The exceptions concerning prescription renewals for chronic medications and physical therapy referrals could be explained by the increased use of teleconsultations in Belgium, which appeared to be an accessible way for patients to consult their primary healthcare provider to some extent [34, 35]. Although chronic medication prescriptions remained stable during the Covid-19 waves and lockdowns, acute medication prescriptions diminished during the Covid-19 waves and lockdowns, which was in line with the decline in the registration of acute diseases, excluding Covid-19. After the first lockdown the different care metrics returned to pre-pandemic levels indicating at least a partial compensation for the reduced primary healthcare use, which corresponds with previous research [33].

Finally, although in general the incidence of chronic diseases showed a decrease during the lockdowns and an increase during the subsequent relaxation periods, some diseases did not follow this trend. Notably, the incidence of chronic lung diseases remained low during the first Covid-19 year, while the incidence of obesity (80%) and chronic cardiovascular conditions (12%) showed a greater increase in the relaxation months after the second lockdown. A potential explanation for the former might be the decrease in air pollution due to the lockdown(s) and the measures that were taken to combat Covid-19 [36, 37]. The latter could be linked to more patients seeking treatment for these conditions or rather stick to the hygienic measures more strictly as these conditions were known risk factors for developing severe Covid-19. In addition, the registration of the body mass index and of certain chronic cardiovascular

conditions is known to be suboptimal, indicating there was greater room for improvement in the registration of these chronic conditions [38–40].

## Strengths and limitations

One of the strengths of this study is the use of a large primary care morbidity registry which allowed us to investigate a sample of the Flemish population. Furthermore, a 'proxy' for the socio-economic status of the patients in Intego was calculated and the correlation of this social determinant with the incidence of chronic diseases during the first year of the Covid-19 syndemic could be assessed. In addition, Intego currently automatically collects data directly from the EHR of the participating GPs on a weekly basis, which provided this study with a robust basis, and increases the potential to further develop as a monitoring instrument.

However, this study also has some limitations. Data is collected from 105 practices. These practices are individually responsible for delivering high quality and complete data. Registration errors and differences in registrations between GPs in the EHR will therefore affect the quality of our results. In addition, since the data is based mainly on physicians' notes in the EHR, additional sources of bias should be considered [41]. For example, it is not clear how circumstantial factors (such as a lockdown) could influence the quality of registration in the EHR. However, the resulting bias is minimized by accounting for practice differences in the analysis and steps have been taken to improve data completeness [42]. We collect data from all patients visiting the participating practices, but this does not guarantee that these patients will be a representative sample for the Flemish population. Finally, a distinction between different types of patient contact (e.g. teleconsultations versus face-to-face consultation) was not possible since this data is not collected in Intego.

## Conclusion

During the first Covid-19 year, overall care provision and the incidence of acute diagnoses increased slightly due to Covid-19 related care, whereas the incidence of chronic diseases was considerably reduced. Obesity was the only chronic disease of which the incidence increased during the first Covid-19 year. On a more granular level, primary care provision and the incidence of chronic diseases decreased in Flanders (Belgium) during the lockdowns in the first Covid-19 year, especially for people with a lower socio-economic status. After the lockdowns in which the hygienic measures were reduced the primary care provision and registration of chronic diseases returned to baseline. This paper strengthened the claim that Covid-19 should be considered as a syndemic instead of a pandemic.

## Supporting information

**S1 Appendix. ICPC codes used in the study.**
(DOCX)

**S2 Appendix. Regression coefficients for the effect of demographic variables on the number of registered health outcomes.**
(DOCX)

**S1 Data.**
(CSV)

**S2 Data.**
(CSV)

## Author Contributions

**Conceptualization:** Steve Van den Bulck, Jonas Crèvecoeur, Bert Vaes.

**Formal analysis:** Jonas Crèvecoeur, Thomas Neyens.

**Project administration:** Steve Van den Bulck.

**Supervision:** Bert Aertgeerts, Bert Vaes.

**Visualization:** Jonas Crèvecoeur.

**Writing – original draft:** Steve Van den Bulck, Jonas Crèvecoeur, Bert Vaes.

**Writing – review & editing:** Steve Van den Bulck, Jonas Crèvecoeur, Bert Aertgeerts, Nicolas Delvaux, Thomas Neyens, Gijs Van Pottelbergh, Patrick Coursier, Bert Vaes.

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
