## [Decision Letter · Decision Letter 0]

8 Mar 2022

PONE-D-21-38141The impact of the Covid-19 pandemic on the incidence of diseases and the provision of primary care: a registry-based studyPLOS ONE

Dear Dr. Van den Bulck,

Thank you for submitting your manuscript to PLOS ONE. After careful consideration, we feel that it has merit but does not fully meet PLOS ONE’s publication criteria as it currently stands. Therefore, we invite you to submit a revised version of the manuscript that addresses the points raised during the review process.

1.- The study design is not appropriate to describe diseases incidences, especially for chronic diseases. If a doctor registered a diagnosis on a date, it is reasonable to assume that she/he has provided attention to the patient for such health problem, so it would be more correct to refer it as attended morbidity or reason for encounter instead of incidence. Please, rewrite the text accordingly.

2. The description of “care provision” needs further explanations. Did the doctors include a new entry in EHRs as a result of a face-to-face contact with the patient, telemedicine, or both? Are there other reasons for entries (for example, revision of laboratory results, prescription refills without patient contact)? Do all entries have to include a diagnosis or it is possible to avoid that information?

3. It should be recognized that the study was exclusively based on physicians’ notes. Especially during lockdown, clinicians could be more concerned in the registry of some health problems than others, but this may not imply a decrease in care. For example, it seems that in such periods doctors refilled chronic prescriptions although they did not record diagnoses. Please, consider such possibility, adding further explanations and citing it as a limitation.

4 . Address all the coments of the reviewers

We look forward to receiving your revised manuscript.

Kind regards,

Juan F. Orueta, MD, PhD

Academic Editor

PLOS ONE

Journal Requirements:

Please, take note of the policy of PLOS ONE on data availability and the STROBE guidelines for observational studies

Additional Editor Comments (if provided):

Numbers of bibliographic references usually are placed before the punctuation mark.

There is a typo on the caption of figure 2. It should be numbered as figure 2 (instead of figure 1)

Reviewers' comments:

Reviewer's Responses to Questions

**Comments to the Author**

1. Is the manuscript technically sound, and do the data support the conclusions?

Reviewer #1: Yes

Reviewer #2: Partly

2. Has the statistical analysis been performed appropriately and rigorously? 

Reviewer #1: Yes

Reviewer #2: Yes

3. Have the authors made all data underlying the findings in their manuscript fully available?

Reviewer #1: No

Reviewer #2: Yes

4. Is the manuscript presented in an intelligible fashion and written in standard English?

Reviewer #1: Yes

Reviewer #2: Yes

5. Review Comments to the Author

Reviewer #1: The subject of the manuscript is of interest to researchers and health professionals. In this sense, various studies have been published in recent months about an increase or, on the contrary, a decrease in acute diagnoses and, above all, a decrease in care for chronic patients during the Covid-19 pandemic.

Comments:

In the introduction it would be of interest to also consider the term syndemic that has been introduced by medical anthropologists to label the synergistic interaction of two or more coexistent diseases and resulting excess burden of disease [Singer, Merrill, and Scott Clair. “Syndemics and Public Health: Reconceptualizing Disease in Bio-Social Context.” Medical Anthropology Quarterly, Vol. 17, no. 4, [American Anthropological Association, Wiley], 2003, p. 423–41, http://www.jstor.org/stable/3655345.]. This aspect is not addressed, however it appears in the conclusions. This term should be well discussed, especially considering its definition and related to the pandemic situation.

In the methodology section, it should be explained more fully what health interventions includes "care provision"

In Table 1, is the diagnosis of stroke included in the diagnoses of cardiovascular disease or neurological disease?

In the discussion, a syndemic situation is neither clearly addressed nor well justified.

Finally, there are several references without any format or style: 9,16,17,18,19,32. In addition, the links do not work in some of these references.

Reviewer #2: The study adds to the growing literature of the impact of the Covid-19 pandemic on health service delivery, by virtue of having access to a large population of health service users in Flanders, Belgium.

My primary concern with the paper is the use of the word ‘incidence’ when discussing changes in patient encounters for chronic disease care. The usual interpretation of the word would be new cases of the disease under consideration, and not just encounters for the disease, such as monitoring or follow-up visits. It appears that the authors mean the latter, which would be an incorrect use of the term, or if they do mean the former, then they should describe how they distinguished new cases of chronic disease from follow-up treatment for the same disease in the same patient. From an intervention point of view, it is important to know whether Covid-19 just reduces access to chronic care in a general way, or whether it actually prevent (or even enhance) the detection of new cases of chronic disease.

Although a shift to the use of telemedicine is postulated as a possible cause of the observed effect in the discussion section, there should be mention in the method section of any effort to detect or quantify alternative routes to care that may have emerged during the pandemic, or indeed showed an increasing trend prior to the pandemic with a resulting possibility of biasing the results.

In the case of acute care, are there any seasonal effects that should be taken into consideration in the interpretation of the pre- and post-pandemic rates of care?

How did the authors in their search strategy for covid-19 related terms, dealt with negation, e.g. ‘not covid’ occurring in the notes they searched to identify such cases?

How representative is the study population of the general Flemish population in terms of age, sex, geography and socio-economic status?

6. PLOS authors have the option to publish the peer review history of their article (what does this mean?). If published, this will include your full peer review and any attached files.

Reviewer #1: No

Reviewer #2: **Yes: **Kobus Herbst

---

## [Author Response · Author response to Decision Letter 0]

25 May 2022

Response to reviewers:

We would like to thank the editor and reviewers for their comments and critical appraisal of the manuscript. We have responded to all comments and have adjusted the manuscript accordingly. Please also see the manuscript with track changes for the adjustments we made.

1.- The study design is not appropriate to describe diseases incidences, especially for chronic diseases. If a doctor registered a diagnosis on a date, it is reasonable to assume that she/he has provided attention to the patient for such health problem, so it would be more correct to refer it as attended morbidity or reason for encounter instead of incidence. Please, rewrite the text accordingly.

Answer (please also see our response to reviewer 2) : We did not provide too much detail on the procedures the Intego registry uses and understand why the editor and reviewers think the study design isn’t appropriate to describe diseases incidence. However, we would like to argue that our study design is nevertheless suitable for describing diseases incidences for of the following reasons:

- We agree with the reviewers/editor that only diseases for which patients consult or seek contact are registered and that this is not identical to diseases incidence. However, because Intego is a large morbidity registry and every diagnosis that is recorded is a new diagnosis (not just encounters for the disease), our ‘observed incidence’ approaches the ‘true incidence’. To illustrate this, there are already a number of publications that describe the use of Intego to measure and analyze diseases incidences (and prevalence). [1-4] To avoid confusion, we have changed the use of the word ‘incidence’ in the method section of the manuscript to ‘observed incidence’

- Intego applies an algorithm that was developed by the Dutch NIVEL institute. This algorithm was developed to construct episodes of illness based on routinely collected EHR data to describe morbidity rates and to differentiate between new cases and chronic cases from follow-up treatment for the same disease in the same patient. [5] Supplemental material 1 contains the list of all ICPC codes and their respective episode of illness. To summarize, based on the NIVEL algorithm the episode of illness starts with the date of diagnosis and ends with the time of the last encounter plus the duration of the contact-free interval. [5]

- For chronic diseases no contact-free interval was defined since they were considered irreversible and there was no need for a contact-free interval. [5]

In order to provide more detail, we have added reference 5 to the paper in the methods section (data). For reference: In order to use Intego data for analyses of the observed diseases’ incidence, an algorithm developed by the Dutch NIVEL institute was used.

We also added the following to the method sections (diagnoses and medication): The aforementioned algorithm, designed by the Dutch NIVEL, enabled a differentiation between new cases and chronic cases from follow-up treatment for the same disease in the same patient.

If the editor/reviewers prefer, we could also add some extra detail and references (1-4 in this rebuttal letter) to illustrate the (past) use of the Intego registry for incidence and prevalence measurements/analyses. 

2. The description of “care provision” needs further explanations. Did the doctors include a new entry in EHRs as a result of a face-to-face contact with the patient, telemedicine, or both? Are there other reasons for entries (for example, revision of laboratory results, prescription refills without patient contact)? Do all entries have to include a diagnosis or it is possible to avoid that information?

Answer: Thank you for pointing out this unclarity. Care provision includes any contact that results in a change in the patient’s medical history including face-to-face contact, telemedicine and prescription refills. We are unable to distinguish these types of contact as this is not registered in the medical history. A diagnosis is not required. We have rewritten the paper to make this clear.

For reference: Care provision includes any action by the GP resulting in a change in the patient’s medical history. This includes face-to-face contact, telemedicine and prescription refills. For a more granular view, we split care provision into separate metrics per record type (diagnoses, lab test, prescription,…).

3. It should be recognized that the study was exclusively based on physicians’ notes. Especially during lockdown, clinicians could be more concerned in the registry of some health problems than others, but this may not imply a decrease in care. For example, it seems that in such periods doctors refilled chronic prescriptions although they did not record diagnoses. Please, consider such possibility, adding further explanations and citing it as a limitation.

Answer: Thank you for this comment, this is indeed a limitation of our research although not only physicians’ notes were used for this study. For example, also data from laboratories (lab test are automatically integrated in the EHR) was used. However, we have added this to the limitations in the manuscript and provided some extra background with an interesting reference that investigated this potential form of bias (and others) when using routinely collected EHR data. [6] 

For reference: In addition, since the data is based mainly on physicians’ notes in the EHR, additional sources of bias should be considered [36]. For example, it is not clear how circumstantial factors (such as a lockdown) could influence the quality of registration in the EHR. 

4 . Address all the comments of the reviewers

Answer: We have addressed all comments of the reviewers and adjusted the manuscript accordingly. 

Journal Requirements:

Answer: We have corrected the output style and used the style templates of Plos One.

Answer: We have added this to the manuscript (methods, ethical approval).

For reference: Written informed consent was obtained from every of the 105 Intego primary care practices.

Answer: We have made available all data used for this research. 

We have added some extra information in the data availability statement (in the editorial manager software). 

For reference: 'Data are available as supplemental material except for the data underlying figure 2. The minimal data set necessary to validate this figure (analysis) contains identifying patient-level data which cannot be suitably de-identified or aggregated and can therefore only be accessed inside a monitored analysis environment. These restrictions were imposed by the Belgian National Information Security Committee: section Social Security and Health. Data requests may be sent to Mr. Roel Heijlen, Data Protection Officer healthdata.be (Sciensano), Roel.Heijlen@sciensano.be , Rue Juliette Wytsmanstraat 14, 1050 Brussels. Interested researchers will need to provide their name, first name, professional organization name, email address, mobile number and the database of interest to request access.'

Reviewer #1: The subject of the manuscript is of interest to researchers and health professionals. In this sense, various studies have been published in recent months about an increase or, on the contrary, a decrease in acute diagnoses and, above all, a decrease in care for chronic patients during the Covid-19 pandemic.

Comments:

In the introduction it would be of interest to also consider the term syndemic that has been introduced by medical anthropologists to label the synergistic interaction of two or more coexistent diseases and resulting excess burden of disease [Singer, Merrill, and Scott Clair. “Syndemics and Public Health: Reconceptualizing Disease in Bio-Social Context.” Medical Anthropology Quarterly, Vol. 17, no. 4, [American Anthropological Association, Wiley], 2003, p. 423–41, http://www.jstor.org/stable/3655345.]. This aspect is not addressed, however it appears in the conclusions. This term should be well discussed, especially considering its definition and related to the pandemic situation.

Answer: Thank you for this suggestion. We have added some extra context in the introduction and discussion to address this aspect. 

For reference:

Introduction: In this context, the Covid-19 pandemic is also called a syndemic by other authors [13]. The concept syndemic is used to label the synergistic interaction of two or more coexistent conditions, biological, economical and/or environmental factors and the resulting excess burden of disease [14, 15].

Discussion: Describing Covid-19 as a syndemic corresponded with previous work that identified different socio-economic factors influencing the variation in mortality and morbidity of Covid-19 [28]. For example, ethnic minority groups with a low income in England and Wales, had a greater chance of being exposed to the virus, to die from it, and to transmit the disease to relatives living in the same house [29]. However, other authors have argued that Covid-19 should not be called a global syndemic, as the different factors that drive the disease are contextual. For instance, the management of Covid-19 differed greatly around the world, and while in the USA the disease could be called syndemic this was not the case in New-Zealand [30]. Our results indicated that in the management of both Covid-19 and chronic diseases, attention should be payed to social determinants of health, even in a Western country with a well-established social security system. [12]. Moreover, these social determinants should also play an important role in the timely access to telemedicine since people with a lower socio-economic status could be hindered by insufficient digital literacy, especially during the Covid-19 syndemic. [31, 32].

In the methodology section, it should be explained more fully what health interventions includes "care provision"

Answer: Thank you for pointing out this unclarity. Care provision includes any contact that results in a change in the patient’s medical history including face-to-face contact, telemedicine and prescription refills. We are unable to distinguish these types of contact as this is not registered in the medical history. A diagnosis is not required. We have rewritten the paper to make this clear.

For reference: Care provision includes any action by the GP resulting in a change in the patient’s medical history. This includes face-to-face contact, telemedicine and prescription refills. For a more granular view, we split care provision into separate metrics per record type (diagnoses, lab test, prescription,…).

In Table 1, is the diagnosis of stroke included in the diagnoses of cardiovascular disease or neurological disease?

Answer: The diagnosis of stroke is included in the diagnoses of cardiovascular disease. For reference please see the appendix with the ICPC codes which states: K90 : Stroke/cerebrovascular accident: Longterm – Chronic : Chronic cardiovascular

In the discussion, a syndemic situation is neither clearly addressed nor well justified.

Answer: We have addressed this issue in the discussion. 

For reference: This latest finding reinforced the claim of other authors that the Covid-19 pandemic should rather be considered as a Covid-19 syndemic. [13]. Describing Covid-19 as a syndemic also corresponded with previous work that identified different socio-economic factors influencing the variation in mortality and morbidity of Covid-19 [28]. For example, ethnic minority groups with a low income in England and Wales, had a greater chance of being exposed to the virus, to die from it, and to transmit the disease to relatives living in the same house [29]. However, other authors have argued that Covid-19 should not be called a global syndemic, as the different factors that drive the disease are contextual [30]. For instance, the management of Covid-19 differed greatly around the world, and while in the USA the disease could be called syndemic this was not the case in New-Zealand [30]. Our results indicated that In Flanders (Belgium) Covid-19 could be consider syndemic and illustrated that In the management of both Covid-19 and chronic diseases, attention should be payed to social determinants of health, even in a Western country with a well-established social security system. [12].].

Finally, there are several references without any format or style: 9,16,17,18,19,32. In addition, the links do not work in some of these references.

Answer: Thank you for pointing this out, we indeed made an error in our reference manager software output style. We have corrected the references and links. All the links work although one URL (reference 37: https://www.eea.europa.eu/highlights/air-pollution-goes-down-as ) does not work when you click on it but does work when you copy/paste it in a browser (Google chrome). It appears this is a problem related to an unsupported protocol that runs on the website that we can not correct. If the reviewer/editor prefers, we could remove this reference.

Reviewer #2: The study adds to the growing literature of the impact of the Covid-19 pandemic on health service delivery, by virtue of having access to a large population of health service users in Flanders, Belgium.

My primary concern with the paper is the use of the word ‘incidence’ when discussing changes in patient encounters for chronic disease care. The usual interpretation of the word would be new cases of the disease under consideration, and not just encounters for the disease, such as monitoring or follow-up visits. It appears that the authors mean the latter, which would be an incorrect use of the term, or if they do mean the former, then they should describe how they distinguished new cases of chronic disease from follow-up treatment for the same disease in the same patient. 

Answer: Thank you for this advice, we used the usual interpretation of the word incidence (new cases of the disease). We did not provide too much detail on the procedures the Intego registry uses and therefore understand why the editor and reviewers think the study design isn’t appropriate to describe diseases incidence. However, we would like to argue our study design is nevertheless suitable for describing diseases incidences for the following reasons:

- We agree with the reviewers/editor that only diseases for which patients consult or seek contact are registered and that this is not identical to diseases incidence. However, because Intego is a large morbidity registry and every diagnosis that is recorded is a new diagnosis (not just encounters for the disease), our ‘observed incidence’ approaches the ‘true incidence’. To illustrate this, there are already a number of publications that describe the use of Intego to measure and analyze diseases incidences (and prevalence). [1-4] To avoid confusion, we have changed the use of the word ‘incidence’ in the method section of the manuscript to ‘observed incidence’

- Intego applies an algorithm that was developed by the Dutch NIVEL institute. This algorithm was developed to construct episodes of illness based on routinely collected EHR data to describe morbidity rates and to differentiate between new cases and chronic cases from follow-up treatment for the same disease in the same patient. [5] Supplemental material 1 contains the list of all ICPC codes and their respective episode of illness. To summarize, based on the NIVEL algorithm the episode of illness starts with the date of diagnosis and ends with the time of the last encounter plus the duration of the contact-free interval. [5]

- For chronic diseases no contact-free interval was defined since they were considered irreversible and there was no need for a contact-free interval. [5]

In order to provide more detail, we have added reference 5 to the paper in the methods section (data). For reference: 

In order to use Intego data for analyses of the observed diseases’ incidence, an algorithm developed by the Dutch NIVEL institute was used.

We also added the following to the method sections (diagnoses and medication): The aforementioned algorithm, designed by the Dutch NIVEL, enabled a differentiation between new cases and chronic cases from follow-up treatment for the same disease in the same patient.

If the editor/reviewers prefer, we could also add some extra detail and references (1-4 in this rebuttal letter) to illustrate the (past) use of the Intego registry for incidence and prevalence measurements/analyses. 

From an intervention point of view, it is important to know whether Covid-19 just reduces access to chronic care in a general way, or whether it actually prevent (or even enhance) the detection of new cases of chronic disease.

Answer: For a number of chronic disease groups we monitored in our paper the number of new diagnoses and compared this to disease incidence in previous years. We observed a sharp decline in the number of recorded new chronic diagnoses in the first pandemic wave. For most chronic diseases it seems unlikely that COVID-19 has in the short-term reduced the actual incidence rate. Hence, the perceived drop-in incidence rate corresponds to a drop in the detection rate.

On the other hand, we also observed that prescriptions for chronic diseases remained on pre-pandemic levels. This shows that essential first line care for chronic patients remained available during the pandemic.

Although a shift to the use of telemedicine is postulated as a possible cause of the observed effect in the discussion section, there should be mention in the method section of any effort to detect or quantify alternative routes to care that may have emerged during the pandemic, or indeed showed an increasing trend prior to the pandemic with a resulting possibility of biasing the results.

Answer: Thank you for this comment. We did not mention this in the method section because we are unable to distinguish between types of contact (e.g. teleconsultation versus face-to-face consultation) as this is not registered in the medical history. We have added this to the limitations. For reference: Finally, a distinction between different types of patient contact (e.g. teleconsultations versus face-to-face consultation) was not possible since this data is not collected in Intego.

In the case of acute care, are there any seasonal effects that should be taken into consideration in the interpretation of the pre- and post-pandemic rates of care?

Answer: We agree that it is important to correct for seasonal variation. Therefore, relative incidences in Figure 1 and Table 1 are computed by comparing the same weeks/months in the first year of the Corona pandemic to these weeks/months in the two previous years. 

How did the authors in their search strategy for covid-19 related terms, dealt with negation, e.g. ‘not covid’ occurring in the notes they searched to identify such cases?

Answer: Thank you for this interesting question. 

We searched for Covid-19 related terms in lab tests and diagnoses. For the lab tests we identify all Covid-19 related lab tests irrespective of the outcome. This is in line with our procedure for other lab tests, where we also include both positive and negative outcomes and mainly demonstrates the surge in lab tests due to the Covid-19 pandemic. 

For the diagnoses, attention values are available in the EHR (active, passive relevant, passive non-relevant and not-present). These attention values are used to remove diagnoses with code ‘not present’. The same filter is applied to diagnoses for other diseases such that only actual diagnoses are retained.

We added this to the method section. For reference: The attention values registered for each diagnosis in the EHR (active, passive relevant, passive non-relevant and not present) were used to omit diagnoses with code ‘not present’. 

How representative is the study population of the general Flemish population in terms of age, sex, geography and socio-economic status?

Answer: The representativeness of the Intego study population is explained in more detail in a previous publication. [7] To illustrate the representativeness of the study population in terms of age and sex, we have made this figure for the reviewers. We did not add this figure to the manuscript to avoid a too lengthy paper, but if the reviewers prefer we could add them to the supplemental material. In Flanders there is 14.08% of the population with a lower socio-economic status while in our study population this percentage is 14.80%.

Additional Editor Comments (if provided):

Numbers of bibliographic references usually are placed before the punctuation mark.

Answer: We have adjusted this.

There is a typo on the caption of figure 2. It should be numbered as figure 2 (instead of figure 1)

Answer: Thank you for your alertness, we have corrected this error.

- [1] Smeets M, Vaes B, Mamouris P, Van Den Akker M, Van Pottelbergh G, Goderis G, et al. Burden of heart failure in Flemish general practices: a registry-based study in the Intego database. BMJ open. 2019; 9(1): e022972.

- [2] Vaes B, Ruelens C, Saikali S, Smets A, Henrard S, Renard F, et al. Estimating the prevalence of diabetes mellitus and thyroid disorders using medication data in Flanders, Belgium. European journal of public health. 2018; 28(1): 193-8.

- [3] Laleman N, Henrard S, van den Akker M, Goderis G, Buntinx F, Van Pottelbergh G, et al. Time trends in statin use and incidence of recurrent cardiovascular events in secondary prevention between 1999 and 2013: a registry-based study. BMC Cardiovasc Disord. 2018; 18(1): 209.

- [4] Spitaels D, Mamouris P, Vaes B, Smeets M, Luyten F, Hermens R, et al. Epidemiology of knee osteoarthritis in general practice: a registry-based study. BMJ open. 2020; 10(1): e031734.

- [5] Nielen MMJ, Spronk I, Davids R, Korevaar JC, Poos R, Hoeymans N, et al. Estimating Morbidity Rates Based on Routine Electronic Health Records in Primary Care: Observational Study. JMIR medical informatics. 2019; 7(3): e11929.

- [6] Verheij AR, Curcin V, Delaney CB, McGilchrist MM. Possible Sources of Bias in Primary Care Electronic Health Record Data Use and Reuse. J Med Internet Res. 2018; 20(5): e185.

- [7] Truyers C, Goderis G, Dewitte H, Akker M, Buntinx F. The Intego database: background, methods and basic results of a Flemish general practice-based continuous morbidity registration project. BMC medical informatics and decision making. 2014; 14: 48.

---

## [Decision Letter · Decision Letter 1]

23 Jun 2022

The impact of the Covid-19 pandemic on the incidence of diseases and the provision of primary care: a registry-based study

PONE-D-21-38141R1

Dear Dr. Van den Bulck,

We’re pleased to inform you that your manuscript has been judged scientifically suitable for publication and will be formally accepted for publication once it meets all outstanding technical requirements.

Kind regards,

Ricardo Jorge Alcobia Granja Rodrigues, Ph.D.

Academic Editor

PLOS ONE

Additional Editor Comments (optional):

You have thoroughly addressed the issues raised in the previous review round, including those pertaining to data availability.

In the editing stage of the manuscript, please double-check for repetition of Figures.

Congratulations on your research.

Reviewers' comments:

Reviewer's Responses to Questions

**Comments to the Author**

1. If the authors have adequately addressed your comments raised in a previous round of review and you feel that this manuscript is now acceptable for publication, you may indicate that here to bypass the “Comments to the Author” section, enter your conflict of interest statement in the “Confidential to Editor” section, and submit your "Accept" recommendation.

Reviewer #1: (No Response)

Reviewer #2: All comments have been addressed

2. Is the manuscript technically sound, and do the data support the conclusions?

Reviewer #1: Yes

Reviewer #2: Yes

3. Has the statistical analysis been performed appropriately and rigorously? 

Reviewer #1: Yes

Reviewer #2: Yes

4. Have the authors made all data underlying the findings in their manuscript fully available?

Reviewer #1: Yes

Reviewer #2: Yes

5. Is the manuscript presented in an intelligible fashion and written in standard English?

Reviewer #1: Yes

Reviewer #2: Yes

6. Review Comments to the Author

Reviewer #1: The authors have considered the observations and recommendations in the first review, despite the limitations of the study. No comments are considered to be added in this second review. The changes made are accepted.

Reviewer #2: The authors have adequately addressed to concerns raised, in particular the use of the term 'incidence'.

7. PLOS authors have the option to publish the peer review history of their article (what does this mean?). If published, this will include your full peer review and any attached files.

Reviewer #1: No

Reviewer #2: **Yes: **Kobus Herbst

---

## [Editor Report · Acceptance letter]

27 Jun 2022

PONE-D-21-38141R1 

The impact of the Covid-19 pandemic on the incidence of diseases and the provision of primary care: a registry-based study 

Dear Dr. Van den Bulck:

I'm pleased to inform you that your manuscript has been deemed suitable for publication in PLOS ONE. Congratulations! Your manuscript is now with our production department. 

Kind regards, 

on behalf of

Dr. Ricardo Jorge Alcobia Granja Rodrigues 

Academic Editor

PLOS ONE